# circRNAs May Be Involved in Dysfunction of Neutrophils of Type 2 Diabetic Mice through Regulation of Specific miRNAs

**DOI:** 10.3390/biomedicines10123129

**Published:** 2022-12-04

**Authors:** Takahiro Umehara, Ryoichi Mori, Kimberly A. Mace, Katsumi Tanaka, Noriho Sakamoto, Kazuya Ikematsu, Hiroaki Sato

**Affiliations:** 1Department of Forensic Medicine, School of Medicine, University of Occupational and Environmental Health, 1-1 Iseigaoka, Yahatanishi-ku, Kitakyushu, Fukuoka 807-8555, Japan; 2Department of Pathology, Graduate School of Biomedical Sciences, School of Medicine, Nagasaki University, 1-12-4 Sakamoto, Nagasaki 852-8523, Japan; 3Division of Cell Matrix Biology and Regenerative Medicine, School of Biological Sciences, Faculty of Biology, Medicine and Health, University of Manchester, Oxford Road, Manchester M13 9PT, UK; 4Department of Plastic and Reconstructive Surgery, Nagasaki University Hospital, 1-7-1 Sakamoto, Nagasaki 852-8501, Japan; 5Department of Respiratory Medicine, Nagasaki University Graduate School of Biomedical Sciences, 1-7-1 Sakamoto, Nagasaki 852-8501, Japan; 6Division of Forensic Pathology and Science, Department of Medical and Dental Sciences, Graduate School of Biomedical Sciences, School of Medicine, Nagasaki University, 1-12-4 Sakamoto, Nagasaki 852-8523, Japan

**Keywords:** diabetic-derived neutrophil, circular RNA, microRNA, inflammation, wound healing, diabetes, microarray

## Abstract

Diabetes is known to delay wound healing, and this delay is attributed to prolonged inflammation. We found that microRNAs (miRNAs) might be involved in the dysfunction of diabetic-derived neutrophils, and dynamics of neutrophil and chronic inflammation might be initiated by miRNA-regulated genes. Moreover, studies of miRNA function in nephropathy have suggested that circular RNAs (circRNAs), which function as sponges of miRNA to regulate their expression, are potential biomarkers and new therapeutic targets for the diagnosis of diabetic nephropathy. Accordingly, to investigate the molecular mechanism of the regulation of inflammation in diabetic-derived neutrophils, we identified circRNAs in diabetic-derived neutrophils obtained from BKS.Cg-*Dock7^m^ +/+ Lepr^db^*/J (*Lepr^db/db^* and *Lepr^db/+^*) mice using microarrays. Neutrophils from pooled bone marrow of three diabetic and three non-diabetic mice were isolated and total RNA was extracted. Microarray analysis was performed using the Arraystar Mouse Circular RNA Array. The results showed that three circRNAs were significantly increased and six circRNAs were significantly decreased in diabetic-derived neutrophils compared with non-diabetic-derived neutrophils. The expressions of some circRNAs in diabetic-derived neutrophils were more than double those in non-diabetic-derived neutrophils. The circRNAs contain binding sites of miRNAs, which were differentially expressed in diabetic-derived neutrophils. Our results suggest that circRNAs may be involved in the regulation of inflammation in diabetic-derived neutrophils.

## 1. Introduction

As population growth continues to increase, the number of diabetic patients is also expected to increase, with estimates reaching up to approximately 300 million by 2030 [1]. Diabetes causes various complications such as neuropathy and diabetic kidney disease. Diabetic nephropathy is a major chronic microvascular complication that occurs in about 25–40% of type 2 diabetes mellitus cases [2]. Moreover, diabetes can delay the healing of wounds, cause infection, and make it more severe, resulting in an intractable ulcer. Approximately 15% of diabetic patients require amputation of a toe, foot or part of a leg in more than 80% of those cases [3]. The development of intractable ulcers involves the functional decline of immune cells from persistent hyperglycemia, vascular insufficiency, and neuropathy [4,5,6]. Research aimed at developing new therapeutic methods and identifying new therapeutic targets for diabetes is being conducted, which can be facilitated by clarifying the molecular mechanisms involved in delayed diabetic wound healing. However, these mechanisms remain largely unknown.

We previously found that diabetic wounds had a prolonged inflammatory phase and impaired transition to the proliferative phase [7], and that the cause of prolonged inflammation was from the dysfunction of diabetic-derived neutrophils rather than the diabetic environment [8]. Furthermore, our results revealed that microRNAs (miRNAs) exhibit differential expression in diabetic-derived neutrophils compared with non-diabetic-derived neutrophils, and that *miR-129-2-3p* overexpression at the wound site of type 2 diabetic mice accelerated wound healing [9]. These results suggest that upregulating *miR-129-2-3p* may be a strategy to improve the dysfunction of diabetic-derived neutrophils. miRNA may thus be a new therapeutic target for diabetic wounds and may be developed into a new treatment method.

Circular RNAs (circRNAs) are key molecules that are involved in the regulation of miRNA expression. circRNAs are synthesized by back splicing and have been shown to function as miRNA sponges and upregulate the expression of mRNAs targeted by miRNAs [10,11,12]. Multiple studies have shown that circRNAs play important roles in the biological processes of various diseases [13]. circSDHC functions as a sponge for miR-127-3p to promote the proliferation and metastasis of renal cell carcinoma [14]. Circ100084 functions as a sponge of miR-23a-5p to regulate IGF2 expression in hepatocellular carcinoma [15]. circDiaph3 regulates vascular smooth muscle cell differentiation, proliferation, and migration in rat [16]. Screening for circRNAs associated with inflammation in peripheral leukocytes of patients with type 2 diabetes mellitus was performed [17], and circRNAs have shown potential applications as disease biomarkers and novel therapeutic targets [18].

We hypothesized that circRNAs might be involved in regulating miRNAs involved in chronic inflammation in diabetic-derived neutrophils. To explore the potential molecular mechanism of inflammatory control in diabetic-derived neutrophils, we screened for circRNA expressions in diabetic-derived neutrophils using microarrays.

## 2. Materials and Methods

### 2.1. Animals

The Animal Care Committee of Nagasaki University and University of Occupational and Environmental Health approved the protocol for this study (approval numbers: 1906191539 and AE22-013). BKS.Cg-*Dock7^m^ +/+ Lepr^db^*/J (*Lepr^db/db^* and *Lepr^db/+^*) mice (5 weeks old) were purchased from Charles River Laboratories (Yokohama, Japan). Animals were housed under a 12/12 h light/dark cycle (light on: 07:00, light off: 19:00) at constant temperature and humidity and allowed free access to food and water. After euthanasia by overanesthesia, male mice at 8 weeks of age were used and were age-matched to controls. To eliminate the effect of hormonal action related to sexual maturation on skin wound healing, we used only male mice.

### 2.2. Microarray Analysis

Bone marrow (BM) was flushed from femurs and tibiae. Neutrophils from the pooled BM of three diabetic (db) and three non-diabetic (non-db) mice were isolated using an anti-Ly-6G microbead kit (Miltenyi Biotec Inc., Bergisch Gladbach, Germany). Microarray analysis was performed on six pools (three pools of three db BM samples and three pools of three non-db BM samples) using Arraystar Mouse Circular RNA Microarray (Arraystar Inc., Rockville, MD, USA), in accordance with the manufacturer’s instructions. Bioinformatic analyses were performed using Miranda and TargetScan (Miranda, Lewis) (Arraystar Inc.). The data discussed in this publication have been deposited in NCBI’s Gene Expression Omnibus and are accessible through GEO Series accession number GSE213297. 

### 2.3. Isolation of Neutrophils, Macrophages, T Cells, and B Cells

Neutrophils from BM of six non-db mice were isolated using a neutrophil isolation kit (Miltenyi Biotec Inc.). Macrophages, T cells, and B cells from BM of six non-db mice were isolated with a Microbead Kit (Miltenyi Biotec Inc.) in accordance with the manufacturer’s instructions. Cells were incubated with anti-CD11b Ab, anti-CD5 Ab, and anti-CD19 Ab to isolate macrophages, T cells, and B cells, respectively [9].

### 2.4. RNA Isolation for Real-Time Quantitative PCR

Neutrophils were isolated from BM of six db and six non-db mice using a neutrophil isolation kit and anti-Ly-6G Microbead kit (Miltenyi Biotec). Total RNA, including miRNA, was extracted using an miRNeasy Mini kit and miRNeasy Micro kit (QIAGEN, Germantown, MD, USA) in accordance with the manufacturer’s instructions. Total RNA was quantified using a NanoDrop™ 2000 Spectrophotometer (Thermo Fisher Scientific, Waltham, MA, USA). RNA samples were stored at −80 °C until use.

### 2.5. cDNA Synthesis for circRNA and microRNA, and Quantitative Real-Time PCR (qRT-PCR)

Complementary DNA (cDNA) was synthesized from RNA (850 ng) using the circRNA cDNA synthesis kit (Applied Biological Materials Inc., Richmond, BC, Canada) in accordance with the manufacturer’s instructions. qRT-PCR was performed in a 20 µL reaction using EmeraldAmp PCR Master Mix (Takara Bio, Kusatsu, Shiga, Japan) and a Thermal Cycler Dice Real Time System (Takara Bio). The amplification mix and thermal cycling conditions were established as described by Panda et al. [19]. Primers for circRNA were designed for the 100 bp sequence before and after the junction sequence by Primer 3 (Table 1). DNA synthesis of circRNAs was performed by Hokkaido System Science Co., Ltd. (Sapporo, Japan). Glyceraldehyde-3-phosphate dehydrogenase (*Gapdh*) was purchased from Takara Bio Inc. 

Total RNA (350 ng) was used as a template for complementary DNA (cDNA) synthesis for miRNA expression analysis using TaqMan MicroRNA Assay (Thermo Fisher Scientific). qRT-PCR was performed using TaqMan MicroRNA Assay for miRNA expression analysis (Thermo Fisher Scientific). Primers (mmu-miR-92a-2-3p (Catalog #: 4427975, Assay ID: 002496), mmu-miR-450b (Catalog #: 4427975, Assay ID: 001962) and U6 snRNA (Catalog #: 4427975, Assay ID: 001973)) were purchased from Thermo Fisher Scientific. The relative quantification of mRNA transcripts and miRNA was performed using the ∆∆Ct method [20]. 

### 2.6. Statistical Analysis

Data are shown as means ± SD. The statistical significance of differences between means was assessed by Mann–Whitney *U* test, unpaired *t*-test, and one-way ANOVA, followed by Tukey’s multiple comparison test and two-way ANOVA, followed by Bonferroni post-tests to compare replicate means (GraphPad Software, San Diego, CA, USA). A *p*-value < 0.05 was considered significant.

## 3. Results

### 3.1. circRNA Expression Is Altered in Neutrophils from Diabetic Mice

Microarray analysis was performed to evaluate circRNA expression and the results identified 206 circRNAs that were expressed more than 1.5-fold higher in diabetic-derived neutrophils compared with non-diabetic-derived neutrophils (Figure 1A, Appendix A). Among the 206 total circRNAs, 3 circRNAs were significantly increased in diabetic-derived neutrophils (Figure 1B, Table 2). We also identified 247 circRNAs that were decreased to two-thirds or less in fold change analysis (Figure 1A, Appendix A). Of the 247 circRNAs, 6 circRNAs were significantly decreased in diabetic-derived neutrophils (Figure 1B, Table 2).

### 3.2. miR-129-2-3p May Be Regulated by circRNAs in Diabetic-Derived Neutrophils

Previous studies showed that the expression of *miR-129-2-3p* was decreased in diabetic-derived neutrophils [9]. We next examined the upregulated circRNAs for miR-129-2-3p binding sites. However, the three circRNAs that were significantly increased in diabetic-derived neutrophils did not possess miR-129-2-3p binding sites. Therefore, we extracted the top seven circRNAs among the total circRNAs with miR-129-2-3p binding sites using Miranda and TargetScan (Figure 2A). The seven identified circRNAs each had one to four miRNA binding sites.

qRT-PCR showed that the expression of *mmu_circRNA_29357* was significantly increased in diabetic-derived neutrophils compared with non-diabetic-derived neutrophils (Figure 2B). The expression of the other circRNAs could not be detected by qRT-PCR. These results, showing that the expression of *miR-129-2-3p* was significantly decreased whereas the expression of *mmu_circRNA_29357* was significantly increased in diabetic-derived neutrophils, suggest that *miR-129-2-3p* may be a target of *mmu_circRNA_29357*. 

### 3.3. The Function of Diabetic-Derived Neutrophils May Be Regulated by miRNA-circRNA Axis other than miR-129-2-3p-circRNA Axis

We next focused on the six circRNAs that were significantly downregulated in diabetic-derived neutrophils and investigated their potential miRNA targets (Table 3). The expression of *mmu-miR-92a-2-5p*, which was identified as a target of *mmu_circRNA_22278*, was significantly increased in diabetic-derived neutrophils (Figure 3A). The expression of *mmu-miR-450b-5p*, which was identified as a target of *mmu_circRNA_001389*, was not changed between diabetic-derived neutrophils and non-diabetic-derived neutrophils (Figure 3B). These results showed that the circRNA–miRNAs may or may not be involved or related to the function of cells from diabetic mice. Further examination of the interactions between these molecules is needed.

To examine the cellular expression of *mmu-miR-92a-2-5p*, we isolated neutrophils, macrophages, B cells, and T cells from BM in non-db mice and examined the expression of *mmu-miR-92a-2-5p* using qRT-PCR. Its expression was significantly increased in T cells and B cells compared with that in other cells (Figure 3C). This suggests that *mmu-miR-92a-2-5p* might be involved in various biological processes, not just inflammation.

## 4. Discussion

circRNA are circular RNA molecules that act as sponges for miRNAs, competing with miRNA response elements, and thereby controlling post-transcriptional regulation of miRNAs [21,22]. Recent studies have demonstrated the utility of miRNAs and circRNAs as new diagnostic markers and therapeutic targets for various diseases such as liver disease and cancer [23,24,25,26,27,28,29,30]. To successfully develop diagnostic markers and therapeutic agents, there is a need to elucidate the molecular mechanisms underlying the specific disease. Many discoveries have been made about the role of miRNAs in diabetic nephropathy, and studies of miRNAs have deepened our understanding of this condition. Although previous studies on diabetic nephropathy had the limitation of being restricted to animal experiments, the findings suggest that the circRNA_0001946–miR671-5p–CDR1 axis is a potential therapeutic target for diabetic nephropathy. Findings on other circRNA–miRNA axes such as circRNA_15698–miR-185 also suggested that miRNA regulation may trigger diabetic nephropathy or be involved in the progression of diabetic nephropathy [31,32,33]. On the other hand, Hsa_circ_0054633 were also shown to be useful as diagnostic markers of pre-diabetes and type 2 diabetes mellitus [34]. In an animal experiment, mmu_circ_0000250 suppressed miR-128-3p expression and increased SIRT1 expression, thereby promoting diabetic wound healing [35]. Other studies reported that circRNAs may exert therapeutic effects via regulation of miRNAs/mRNAs such as STAT3 and the miR-31–FBN1 axis in wound healing [36,37]. However, to the best of our knowledge, no studies have focused on circRNA expression in neutrophils as a factor associated with delayed wound healing in diabetes mellitus. Accordingly, in this study, we performed a comprehensive analysis of circRNAs in diabetic-derived neutrophils.

Wound healing in diabetic patients is known to be delayed, which can result in intractable skin ulcers [9]. Repair and regeneration of wound skin tissue consists of three phases: inflammation, proliferation/migration, and maturation/resolution. A balance between the appropriate recruitment and elimination of cells involved in inflammation, such as neutrophils and macrophages, from the BM to the wound site is important for effective tissue repair. We previously showed that, although myeloid cells, including granulocytes, gradually converge on wound sites and then disappear 7 days after wounding in non-diabetic mice, in diabetic mice the number of myeloid cells in wound sites continued to increase on days 2 to 10 after wounding [7]. Thus, the accumulation of chronic myeloid cells at wound sites, involving prolongation of the inflammatory phase, impedes the transition to the proliferative phase such as angiogenesis. In diabetic-derived neutrophils, many miRNAs are differentially expressed, and our results identified *miR-129-2-3p* as a downregulated miRNA. *miR-129-2-3p* directly regulates the translation of certain genes such as *Casp6*, *Ccr2*, and *Dedd2* and is involved in various biological processes such as inflammatory responses and apoptosis [9]. We thus speculated that low-level expression of *miR-129-2-3p* may contribute to the dysfunction of diabetic-derived neutrophils. As a factor for the low expression of *miR-129-2-3p* in diabetic-derived neutrophils, other recent studies have shown that miR-129-2 is epigenetically regulated by DNA methylation [38]. In our previous study, analysis of ChIP data of the regulatory region in putative intron 1 of the gene, which is upstream from the sequence encoding the mature miRNA, showed that this region is bound by many transcription factors such as Pu.1 and Cebp transcription factors. Both of these genes are underexpressed in diabetic-derived Gr-1+CD11b+ myeloid cells [39], which include neutrophils. In this study, we speculated that *miR-129-2-3p* is regulated by circRNAs and performed circRNAs screening in diabetic-derived neutrophils. A comprehensive analysis of circRNAs in diabetic-derived neutrophils has not been performed so far. We also did not find any studies linking the nine circRNAs whose expression was significantly changed in this study with wound healing. Of the seven circRNAs with miR-129-2-3p binding sites, only circRNA_29357 showed significant expression changes in diabetic-derived neutrophils by qRT-PCR. Accordingly, further examination of the function of circRNA_29357 in diabetic-derived neutrophils is required.

We performed prediction analysis and identified miR-450b-5p as a potential target miRNA for circRNA_001389 and miR-92a-2-5p as a potential target microRNA for circRNA_22278. miR-450b-5p was previously reported to be associated with inflammation and shown to reverse endothelial cell damage through regulation of circRNA_0005699 [40]. In this study, no significant changes in miR-450b-5p expression were observed in diabetic-derived neutrophils, suggesting that miR-450b-5p may not be involved in the dysfunction of diabetic-derived neutrophils. In contrast, miR-92a-2-5p showed significant expression changes in diabetic-derived neutrophils. miR-92a-2-5p was reported to induce apoptosis by post-transcriptional repression of miR-92a-2-5p through regulation of circAMOTL1L in renal cell carcinoma [41]. Diabetic wounds in type 2 diabetic mice have myeloid cell accumulation in the wound area, and these cells cause abnormal apoptosis. Consequently, these results showed that these causes were caused by an increase in *mmu-miR-92a-2-5p*, suggesting that *mmu_circRNA_22278* may be involved in regulation of apoptosis.

## 5. Conclusions

Our results showed that multiple circRNAs exhibited altered expressions in diabetic-derived neutrophils. circRNAs function as sponges for many miRNAs and affect downstream mRNA expression, suggesting that the aberrant levels of these circRNAs may be involved in the dysfunction of diabetic-derived neutrophils. Further investigation is required to determine whether the identified circRNAs are involved in the function of diabetic-derived neutrophils.

## Figures and Tables

**Figure 1 biomedicines-10-03129-f001:**
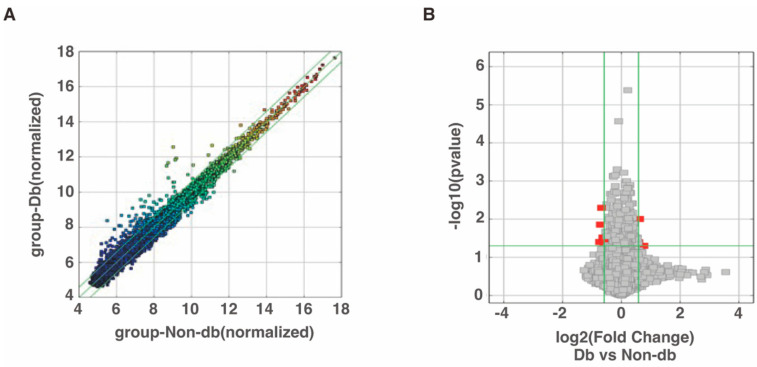
Changes in circRNA expressions in diabetic-derived neutrophils. (**A**) Scatter plot was used for assessing the circRNA expression variation between two samples or two groups of samples. The X- and Y-axes are the normalized signal values of the samples (log2 scaled). The green lines indicate fold change. CircRNAs above the top green line and below the bottom green line are circRNAs with more than 1.5-fold change between the two compared samples. (**B**) Volcano plots were used for visualizing differential expression between two different conditions. The vertical lines correspond to 1.5-fold increase and decrease, respectively, and the horizontal line represents a *p*-value of 0.05. The red points represent the differentially expressed circRNAs with statistical significance.

**Figure 2 biomedicines-10-03129-f002:**
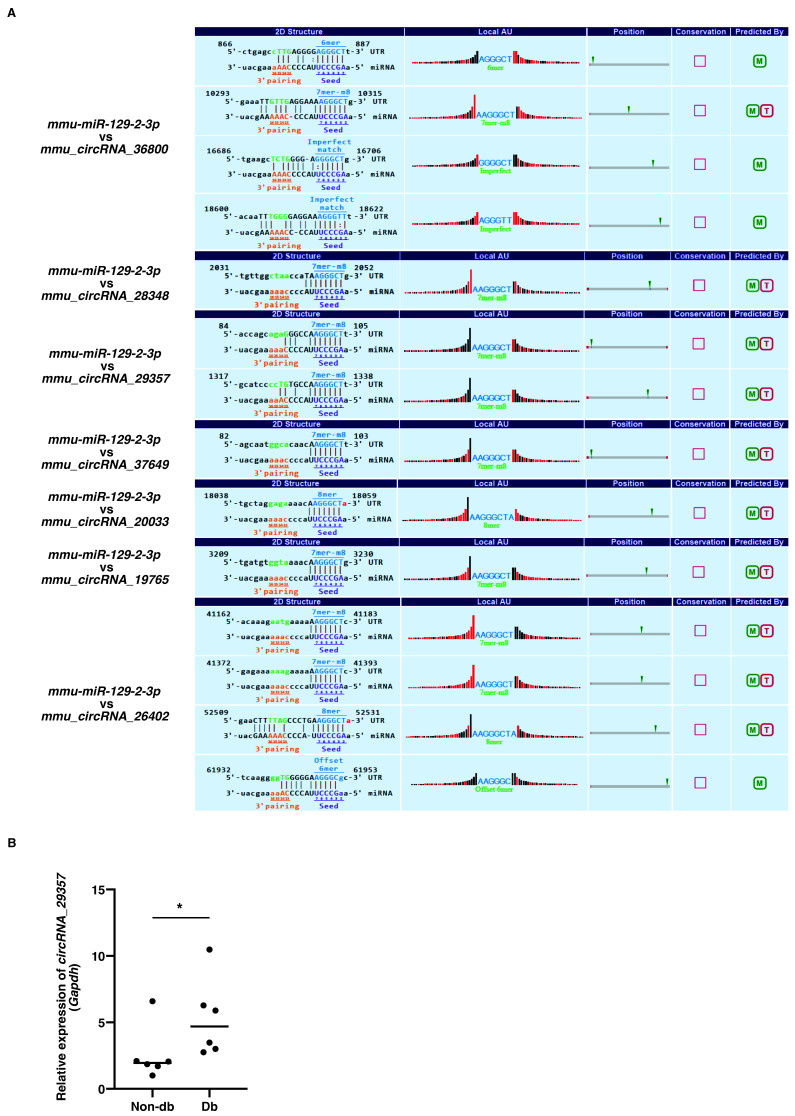
Identification of circRNAs with putative miR-129-2-3p binding sites. (**A**). The top seven circRNAs with putative miR-129-2-3p binding sites using Miranda and TargetScan from 206 circRNAs were extracted. The binding sequences between miR-129-2-3p and circRNAs are shown. (**B**). Relative expression of *circRNA_29357* in Db. Graphs show mean ± SD (*n* = 6). The statistical significance of differences between means was assessed by Mann–Whitney *U* test. * *p* < 0.05.

**Figure 3 biomedicines-10-03129-f003:**
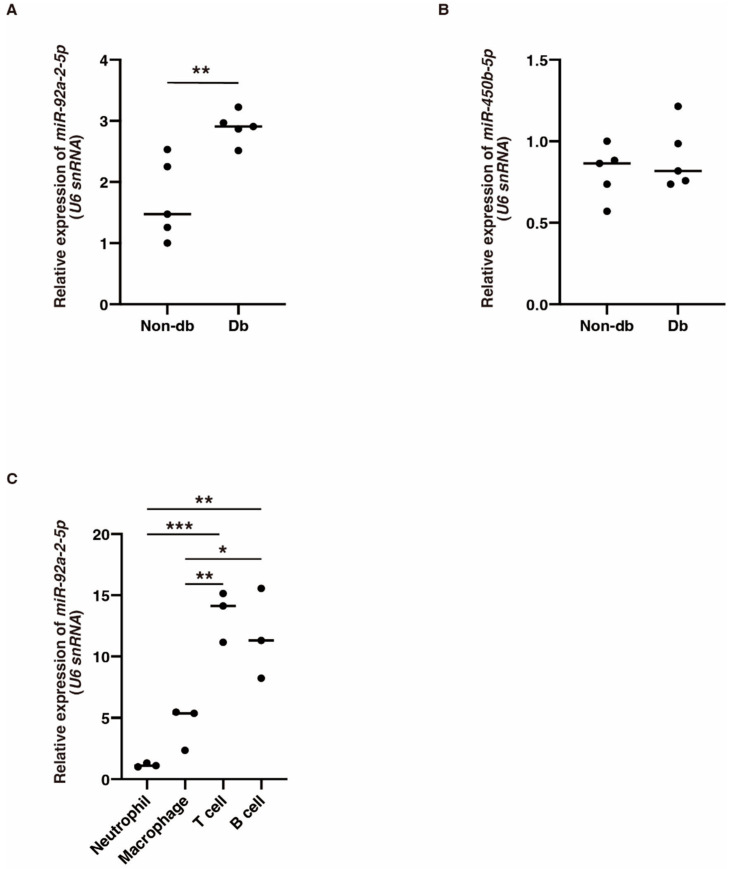
The expression of *miR-92a-2-5p* and *miR-450b-5p* in diabetic-derived neutrophils and *miR-92a-2-5p* in T and B cells. (**A**,**B**). Relative expression of *miR-92a-2-5p* and *miR-450b-5p* in Db. Graphs show mean ± SD (*n* = 5). The statistical significance of differences between means was assessed by unpaired *t*-test. ** *p* < 0.01. (**C**). Relative expression of *miR-92a-2-5p* in neutrophils, macrophages, T cells, and B cells isolated from BM from non-db mice. *miR-92a-2-5p* is mainly expressed in T cells and B cells. Graphs show mean ± SD (*n* = 3). The statistical significance of differences between means was assessed by one-way ANOVA, followed by Tukey’s multiple comparison test. * *p* < 0.05; ** *p* < 0.01; *** *p* < 0.001.

**Table 1 biomedicines-10-03129-t001:** Primers of circRNAs.

Target gene	Primer	Sequence (5′-3′)	Base
mmu_circRNA_36800	forward	GGGCGCTTTCTAGAAGAAGT	20
	reverse	CCAGAAGAATCATAAGCACATGG	23
mmu_circRNA_28348	forward	TAGTTCACCGGCGGATTTAC	20
	reverse	TCTCAGGTGTTCCTTCTGACC	21
mmu_circRNA_29357	forward	TTGCCTGTGATGAGTGTGGT	20
	reverse	TGGTTCAGCTGTAGCAGGAG	20
mmu_circRNA_37649	forward	AGGTGCTGGTCCTACAGAGG	20
	reverse	TCTGTCCAGAAGCCCTTGTT	20
mmu_circRNA_20033	forward	AGCCAATATAGCCTGGATGG	20
	reverse	CACTTCATGAGAACGGCTGA	20
mmu_circRNA_19765	forward	CCCTCGCCAAACATTTTTAT	20
	reverse	GAGTGCAAAGGAAATGCCATA	21
mmu_circRNA_26402	forward	CAGCAGCTGGAAAAGGATCT	20
	reverse	ATTTCTTCCATGGCAGCTTG	20

**Table 2 biomedicines-10-03129-t002:** Differentially expressed circRNAs.

circRNA	Source	Chrom	circRNA_Type	GeneSymbol	FC (abs)	Regulation	*p*-Value
mmu_circRNA_015487	circBase	chr7	exonic	Tead2	1.5407217	up	0.00990479
mmu_circRNA_41782	25714049	chr7	exonic	Hdgfrp3	1.7063881	up	0.04996231
mmu_circRNA_43177	25714049	chr8	exonic	Elmod2	1.5117275	up	0.04014974
mmu_circRNA_001389	circBase	chr19	antisense	Malat1	1.5153945	down	0.04172861
mmu_circRNA_34763	25714049	chr2	exonic	Syndig1	1.5506074	down	0.030088466
mmu_circRNA_41665	25714049	chr7	exonic	Chd2	1.6456881	down	0.013881577
mmu_circRNA_19188	25070500	chr17	intronic		1.6756403	down	0.039428327
mmu_circRNA_22278	25714049	chr10	exonic	Dot1l	1.5997135	down	0.005037364
mmu_circRNA_28776	25714049	chr15	sense overlapping	Mroh4	1.5055119	down	0.034069477

**Table 3 biomedicines-10-03129-t003:** miRNA binding sites of circRNAs.

circRNA	MRE1	MRE2	MRE3	MRE4	MRE5
mmu_circRNA_001389	mmu-miR-7085-3p	mmu-miR-450b-5p	mmu-miR-7664-3p	mmu-miR-6982-3p	mmu-miR-1948-3p
mmu_circRNA_34763	mmu-miR-7656-3p	mmu-miR-6926-3p	mmu-miR-3077-3p	mmu-miR-7033-5p	mmu-miR-7017-3p
mmu_circRNA_41665	mmu-miR-6932-3p	mmu-miR-6946-3p	mmu-miR-6908-3p	mmu-miR-6961-3p	mmu-miR-6957-3p
mmu_circRNA_19188	mmu-miR-5110	mmu-miR-6953-5p	mmu-miR-1249-5p	mmu-miR-6981-5p	mmu-miR-346-3p
mmu_circRNA_22278	mmu-miR-6931-5p	mmu-miR-7048-5p	mmu-miR-7682-3p	mmu-miR-92a-2-5p	mmu-miR-7036b-5p
mmu_circRNA_28776	mmu-miR-6896-3p	mmu-miR-7000-5p	mmu-miR-5615-5p	mmu-miR-6989-5p	mmu-miR-6987-5p

## Data Availability

Publicly available datasets were analyzed in this study. These data can be found here: GSE213297.

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
