# Peer review of "circRNAs May Be Involved in Dysfunction of Neutrophils of Type 2 Diabetic Mice through Regulation of Specific miRNAs"

_biomedicines, 2022, doi:10.3390/biomedicines10123129_

Round 1
Reviewer 1 Report
The manuscript entitled "circRNAs may be involved in dysfunction of neutrophils of type 2 diabetic mice through regulation of specific miRNAs" by Takahiro et al., identifies microRNAs that might be involved in the dysfunction of diabetic-derived neutrophils. The authors identified three circRNAs with significantly increased expression and 6 circRNAs were significantly decreased in diabetic derived neutrophils. The authors towards the later half of the manuscript speculate that the circRNAs contain binding sites of miRNAs, which were differentially expressed in diabetic derived neutrophils. However, the authors do not establish any concrete evidence towards the presence of such miRNAs with the cicrRNAs. Further experiments in this direction in the likes of cross linking and pulldowns would help support the authors claims. In addition, the manuscript would gain from further validation through gain of function or loss of function studies with respect to the top hits found from the study. I strongly feel that this manuscript needs more experimental validation before concluding something strong. I believe the manuscript can be considered if additional experiments are added in this regard.
Author Response
Point 1: The manuscript entitled "circRNAs may be involved in dysfunction of neutrophils of type 2 diabetic mice through regulation of specific miRNAs" by Takahiro et al., identifies microRNAs that might be involved in the dysfunction of diabetic-derived neutrophils. The authors identified three circRNAs with significantly increased expression and 6 circRNAs were significantly decreased in diabetic derived neutrophils. The authors towards the later half of the manuscript speculate that the circRNAs contain binding sites of miRNAs, which were differentially expressed in diabetic derived neutrophils. However, the authors do not establish any concrete evidence towards the presence of such miRNAs with the cicrRNAs. Further experiments in this direction in the likes of cross linking and pulldowns would help support the authors claims. In addition, the manuscript would gain from further validation through gain of function or loss of function studies with respect to the top hits found from the study. I strongly feel that this manuscript needs more experimental validation before concluding something strong. I believe the manuscript can be considered if additional experiments are added in this regard.
Response 1: As you pointed out, the conclusions of this study are highly speculative, but it is clear from previous studies that circRNAs influence various diseases by affecting the regulation of miRNAs and mRNAs. Therefore, it is suspected that they are involved in the dysfunction of diabetic-derived neutrophils. By publishing the data from the comprehensive analysis of circRNAs performed in this study in this journal, which has a wide readership, we would like to provide many researchers with the opportunity to verify our findings.
As you pointed out, this research requires additional experiments, but owing to the effects of COVID-19, it will take several months to obtain reagents, antibodies, and so on in order to conduct such experiments. Considering this, we decided to submit the manuscript at the current stage. We also consider the importance of making this paper available as soon as possible as a resource for other researchers.
Reviewer 2 Report
In this interesting article the authors aimed to investigate the molecular mechanism of the regulation of inflammation in diabetic-derived neutrophils, through the analysis of circRNAs in diabetic-derived neutrophils.
I suggest you to modify the abstract including: Background/Methods/Results/Conclusion.
The experiment is adequately performed, the statistical analysis sufficient and the data particularly interesting.
I suggest you to underline the possible use of circRNAs in humans and the actual use of miRNAs in DKD patients. I suggest you the paper “Tu C, Wang L, Wei L, Jiang Z. The role of circular RNA in Diabetic Nephropathy. Int J Med Sci. 2022 May 20;19(5):916-923. doi: 10.7150/ijms.71648. PMID: 35693742; PMCID: PMC9149631”, “Zahari Sham SY, Abdullah M, Osman M, Seow HF. An insight of dysregulation of microRNAs in the pathogenesis of diabetic kidney disease. Malays J Pathol. 2022 Aug;44(2):187-201. PMID: 36043582” and for the introduction “Amatruda M, Gembillo G, Giuffrida AE, Santoro D, Conti G. The Aggressive Diabetic Kidney Disease in Youth-Onset Type 2 Diabetes: Pathogenetic Mechanisms and Potential Therapies. Medicina (Kaunas). 2021 Aug 25;57(9):868. doi: 10.3390/medicina57090868. PMID: 34577791; PMCID: PMC8467670”. English language and style are fine/minor spell check required.
This paper can give a relevant contribution to the knowledge of diabetic kidney disease, I support the publication of the study after few improvements.
Author Response
Point 1: In this interesting article the authors aimed to investigate the molecular mechanism of the regulation of inflammation in diabetic-derived neutrophils, through the analysis of circRNAs in diabetic-derived neutrophils. I suggest you to modify the abstract including: Background/Methods/Results/Conclusion.
Response 1: We have made changes to the abstract (page 1, lines 17–18; page 1, lines 20–23) in accordance with this comment. 
The experiment is adequately performed, the statistical analysis sufficient and the data particularly interesting.
Point 2: I suggest you to underline the possible use of circRNAs in humans and the actual use of miRNAs in DKD patients. I suggest you the paper “Tu C, Wang L, Wei L, Jiang Z. The role of circular RNA in Diabetic Nephropathy. Int J Med Sci. 2022 May 20;19(5):916-923. doi: 10.7150/ijms.71648. PMID: 35693742; PMCID: PMC9149631”, “Zahari Sham SY, Abdullah M, Osman M, Seow HF. An insight of dysregulation of microRNAs in the pathogenesis of diabetic kidney disease. Malays J Pathol. 2022 Aug;44(2):187-201. PMID: 36043582” and for the introduction “Amatruda M, Gembillo G, Giuffrida AE, Santoro D, Conti G. The Aggressive Diabetic Kidney Disease in Youth-Onset Type 2 Diabetes: Pathogenetic Mechanisms and Potential Therapies. Medicina (Kaunas). 2021 Aug 25;57(9):868. doi: 10.3390/medicina57090868. PMID: 34577791; PMCID: PMC8467670”. English language and style are fine/minor spell check required.
Response 2: We have made changes to the introduction, discussion, and references (page 1, lines 41–43; page 2, lines 48–51, lines 60–61; pages 8–9, lines 210–221, lines 222–229; page 10, lines 303–304; page 11, lines 366–373; page 12, lines 376–378, line 384) in accordance with this comment.
This paper can give a relevant contribution to the knowledge of diabetic kidney disease, I support the publication of the study after few improvements.
Round 2
Reviewer 1 Report
As mentioned earlier the manuscript is highly speculative and has no validations what so ever. As the authors mentioned that the experiments would take a substantial amount of time for them to complete. In that case the manuscript could be deemed as a source of important circRNA datasets involved in the dysfunction of neutrophils of type 2 diabetic mice.